# Blend of Essential Oils Supplemented Alone or Combined with Exogenous Amylase Compared with Virginiamycin Supplementation on Finishing Lambs: Performance, Dietary Energetics, Carcass Traits, and Nutrient Digestion

**DOI:** 10.3390/ani11082390

**Published:** 2021-08-13

**Authors:** Alfredo Estrada-Angulo, Yesica J. Arteaga-Wences, Beatriz I. Castro-Pérez, Jesús D. Urías-Estrada, Soila Gaxiola-Camacho, Claudio Angulo-Montoya, Elizama Ponce-Barraza, Alberto Barreras, Luis Corona, Richard A. Zinn, José B. Leyva-Morales, Xiomara P. Perea-Domínguez, Alejandro Plascencia

**Affiliations:** 1Faculty of Veterinary Medicine and Zootechnics, Autonomous University of Sinaloa, Culiacan 80260, Mexico; alfred_vet@hotmail.com (A.E.-A.); arteaga.yesi.92@hotmail.com (Y.J.A.-W.); laisa_29@hotmail.com (B.I.C.-P.); uriasestrada_jd@hotmail.com (J.D.U.-E.); soilagaxiola@uas.edu.mx (S.G.-C.); c.angulom@hotmail.com (C.A.-M.); eli_j13@hotmail.com (E.P.-B.); 2Veterinary Science Research Institute, Autonomous University of Baja California, Mexicali 21100, Mexico; beto_barreras@yahoo.com; 3Faculty of Veterinary Medicine and Zootechnics, National Autonomous University of Mexico, México 04510, Mexico; gochi@servidor.unam.mx; 4Animal Science Department, University of California, Davis, CA 95616, USA; razinn@ucdavis.edu; 5Health Sciences Department, University Autonomous of the West, Guasave 81048, Mexico; jose.leyva@uadeo.mx; 6Natural and Exact Sciences Department, University Autonomous of the West, Guasave 81048, Mexico; xiomara.perea@uadeo.mx

**Keywords:** lambs, essential oils, exogenous amylase, virginiamycin, growth performance, digestion

## Abstract

**Simple Summary:**

Antibiotics have been extensively used as growth promoters in livestock, but current interests are focused on limiting the use of conventional antibiotics as feed additives in livestock production. Essential oil compounds belong to a “generally-recognized-as-safe” category of feed additives that may serve as alternatives to conventional antibiotics used as growth promoters. In this study, dietary supplementation of finishing lambs with essential oils alone, or combined with exogenous enzymes, improved dietary energy utilization and meat production in a manner comparable to that of the antibiotic virginiamycin.

**Abstract:**

Two experiments were conducted to compare a supplemental blend of essential oils alone (EO) or combined with enzymes (EO + ENZ) versus virginiamycin (VM), on characteristics of growth performance (Exp. 1) and digestion (Exp. 2) in finishing lambs. Lambs were fed a high-energy finishing diet supplemented with: (1) no supplement (control); (2) 150 mg supplemental EO; (3) 150 mg supplemental EO plus 560 mg alpha-amylase (EO + ENZ); and 4) 25 mg VM. Compared with the control, growth performance response to EO and VM were similar, enhancing (5.7%, *p* < 0.05) feed efficiency and observed dietary net energy. Compared with control, supplementation with EO + ENZ tended (*p* = 0.09) to increase dry matter intake (6.8%), improving (*p* < 0.05) weight gain and feed efficiency (10.4 and 4.4%, respectively). Dietary energy utilization was greater (2.7%, *p* < 0.05) for EO and VM than EO + ENZ. Treatment effects on the carcass and visceral mass were small, but additive supplementation decreased (*p* ≤ 0.03) the relative weight of the intestines. There were no treatment effects on measures of digestion nor digestible energy of the diet. Supplemental EO may be an effective alternative to VM in high-energy finishing diets for feedlot lambs. Combination EO + ENZ may further enhance dry matter intake, promoting increased weight gain.

## 1. Introduction

Virginiamycin (VM) is an antimicrobial (peptolide antibiotic) that inhibits growth of Gram-positive bacteria [1]. At supplementation levels of 22 to 28 mg/kg diet DM, VM enhances average daily gain (ADG), feed efficiency (gain-to-feed ratio, GF), and diet energy utilization in feedlot cattle [2,3]. As well, it reduces the incidence of liver abscess and ruminal lactate accumulation [4]. In a meta-analysis, comparing VM with the ionophore monensin (MON), Gorocica and Tedeschi [5] observed that whereas VM resulted in greater ADG than MON, both additives resulted in similar enhancements in feed efficiency. Current interests in limiting the use of conventional antibiotics as feed additives in livestock production, has led to the search for “generally-recognized-as-safe” additive alternatives. Previous reports [6,7,8] indicate that mixtures of essential oils may result in similar or even greater enhancements in growth performance than MON. These effects may not be attributable to changes in digestion, as the effects of supplemental essential oils on total tract digestion are not appreciable [9,10]. Comparisons of essential oils vs. VM in ruminants is limited. In poultry, growth performance response and gastrointestinal tract health were similar for supplemental peppermint oil vs. VM [11]. Due to its high starch content, corn grain is extensively used as a feed energy source for livestock. Most of the starch in corn grain is within endosperm of the kernel in a granular form; the tight intermolecular bonding between starch molecules, along with the compact nature of the starch granule, impedes rapid moisture uptake (rehydration) and thus, ruminal starch digestion [12]. Because of its hydrolytic action, supplemental α-amylase may increase the availability of starch hydrolysis products in the rumen [13]. Therefore, exogenous α-amylase supplementation may enhance ruminal digestion of starch in cracked corn-based diets [14]. The combination of EO with exogenous enzyme α-amylase has enhanced both DMI and ADG of feedlot cattle that were fed with finishing cracked corn-based diets [15]. We hypothesized that mixture of essential oils may enhance growth performance, dietary energetics, and carcass characteristics of finishing feedlot lambs in a manner comparable to that of virginiamycin, and that the combination of essential oils with exogenous α-amylase might further potentiate that effect. Accordingly, two experiments were conducted to compare the effects of supplemental essential oils alone or combined with enzymes versus virginiamycin on growth performance, dietary net energy, carcass characteristics, visceral mass (Exp. 1) and measures of total tract digestion (Exp. 2). 

## 2. Materials and Methods

This experiment was conducted at the Universidad Autónoma de Sinaloa Feedlot Lamb Research Unit, located in Culiacán, México (24°46′13″ N and 107°21′14″ W). Culiacán is about 55 m above sea level, and has a tropical climate. During the course of the experiment, ambient air temperature averaged 31.5 °C (minimum and maximum of 26.1 and 34.5 °C, respectively), and relative humidity averaged 36.0% (minimum and maximum of 29.7 and 52.8%, respectively). All animal management procedures were conducted within the guidelines of federal-locally-approved techniques for animal use and care [16] and approved by the Ethics Committee of Faculty of Veterinary Medicine and Zootechnics from the Autonomous University of Sinaloa (Protocol #1422019). 

### 2.1. Exp. 1. Growth Performance and Carcass Traits

#### 2.1.1. Animal, Diet, Treatments, and Samples Analyses

Forty-eight Pelibuey × Katahdin crossbred intact male lambs (27.87 ± 4.71 kg initial live weight (LW)) were used in an 87-d experiment to evaluate treatment effects on growth performance and carcass characteristics. Two weeks before initiation of the experiment lambs were treated for parasites (7.5 mg/kg LW; Closantel Panavet 15%, Panamericana Veterinaria de México City, México), injected with 2 mL vitamin A (500,000 UI, 75,000 IU vitamin D_3_, and 50 IU vitamin E; Synt-ADE^®^, Zoetis México, México City), and vaccinated for Mannheimia haemolityca (One Shot Ultra, Zoetis México, México City). Upon initiation of the experiment, lambs were weighed before the morning meal (electronic scale; TORREY TIL/S: 107 2691, TOR REY Electronics Inc, Houston, TX, USA). Lambs were blocked by initial weight (6 blocks) and assigned to 24 pens, two lambs/pen. Dietary treatments were randomly assigned to pens within blocks, 6 replicas per treatment. Pens have 6 m^2^ with overhead shade, automatic waterers and 1 m fence-line feed bunks. Composition of the cracked corn-based basal diet is shown in Table 1. Corn grain was prepared by passing whole regional white corn through rollers (cylinder rollers of 46 cm diameter × 61 cm length, 5.5 corrugations/cm; Memco, Mills Rolls, Mill Engineering & Machinery Co., Oklahoma, CA, USA). Roll pressure was adjusted so that the kernels were broken to produce a bulk density of approximately 0.60 kg/L. Sudangrass hay was ground in a hammer mill (Azteca 20, Molinos Azteca, Guadalajara, México) with a 3.81 cm screen before incorporation into total mixed ration. Treatments consisted of basal diet supplemented with: (1) No additives (control); (2) 150 mg/d of a standardized source of a mixture of essential oils (EO); (3) 150 mg/d EO plus 560 mg/d alpha-amylase (EO + ENZ); and (4) 25 mg/d virginiamycin (VM; Stafac 500, Phibro Animal Health, Ridgefield Park, NJ, USA). The blend of essential oils used contains thymol, eugenol, limonene and vanillin on an organic carrier (CRINA-Ruminants, DSM Nutritional Products, Basel, Switzerland), and the exogenous α-amylase used was produced by *Bacillus licheniformis* (Ronozyme RumiStar, DSM Nutritional Products, Basel, Switzerland). The daily dose of 150 mg EO used was chosen based on a previous report where ingestion of 100 to 200 mg EO/d resulted in maximal enhancements on ruminal fermentation and feed efficiency in lactating ewes [17], and improved feed efficiency and observed dietary net energy in feedlot lambs [8]. The dose of 560 mg ENZ/day was estimated from data publishing from Meschiatti et al. [15]. Level of VM was based on recommended drug label dosage. The treatments (complete mixed diets) were prepared using a 2.5 m^3^ capacity paddle mixer (model 30910-7, Coyoacán, México). To avoid contamination between treatments, the mixer was thoroughly cleaned between each elaborated batch. To ensure additive consumption, the total daily dosage per lamb was mixed in 300 g of basal diet provided in the morning feeding (all lambs were fed the basal control diet in the afternoon feeding). Thus, lambs were provided fresh feed twice daily at 800 and 1400 h. Whereas the amount of feed provided in the morning feeding was constant, feed offered in the afternoon feeding was adjusted daily, allowing for a feed residual ~50 g/kg daily feed offering. Residual feed was collected between 0740 and 0750 h each morning and weighed. The adjustments to either increase or decrease daily feed delivery were provided in the afternoon feeding. Water consumption was measured daily at 700 h by dipping a graduated rod into the tank drinker (one watering tank for each pen). Once the measure was taken, the remaining water was drained, and the tanks were refilled with fresh water. Lambs were weighed just prior to the morning feeding on days 1 and 87 (final day). Live weights (LW) on day 1 was converted to shrunk body weight (SBW) by multiplying LW by 0.96 to adjust for the gastrointestinal fill [18]. All lambs were fasted for 18 h before recording the final LW.

Feed samples were collected for each elaborated batch. Feed refusal was collected daily and composited weekly for DM analysis (oven drying at 105 °C until no further weight loss; method 930.15) [20]. Feed samples were subjected to the following analyses: DM (oven drying at 105 °C until no further weight loss; method 930.15); CP (N × 6.25, method 984.13) according to AOAC [20]; and neutral detergent fiber (NDF) following procedures described by Van Soest et al. [21] (corrected for NDF-ash, incorporating heat stable α-amylase using Ankom Technology, Macedon, NY, USA).

#### 2.1.2. Calculations

Estimates of ADG, and dietary net energy are based on initial SBW and final (d 87) fasted SBW. Average daily gain was computed by subtracting initial SBW from final SBW and dividing the result by the number of days on feed. Feed efficiency was computed as ADG/ daily DMI. One approach for evaluation of the efficiency of dietary energy utilization in growth performance trials is the ratio of observed-to-expected DMI and observed-to-expected dietary NE. Based on diet NE concentration and measures of growth performance, there is an expected energy intake. This estimation of expected DMI is performed based on observed ADG, average SBW, and NE values of the diet (Table 1): expected DMI, kg/d = (EM/NE_m_) + (EG/NE_g_), where EM (energy required for maintenance, Mcal/d) = 0.056 × SBW^0.75^, EG (energy gain, Mcal/d) = 0.276 × ADG × SBW^0.75^, and NE_m_ and NE_g_ are corresponding NE values based on the ingredient composition [19] of the experimental diet (Table 1). The coefficient (0.276) was taken from NRC [22] assuming a mature weight of 113 kg for Pelibuey × Katahdin male lambs [23]. The observed dietary net energy was calculated using EM and EG values, and DMI observed during experiment by means of the quadratic formula:$$x=\frac{-b\pm \sqrt{b^{2}-4ac}}{2c}$$
where *x* = NE_m_, Mcal/kg, *a* = −0.41 EM, *b* = 0.877 EM + 0.41 DMI + EG, and *c* = −0.877 DMI [24].

#### 2.1.3. Carcass Characteristics, Whole Cuts, and Tissue Shoulder Composition

All lambs were harvested on the same day. Lambs were stunned (captive bolt), exsanguinated and skinned. Gastrointestinal organs were separated and weighed, the omental and mesenteric fat were weighed, and hot carcass weight (HCW) was registered. After carcasses (with kidneys and internal fat included) chilled in a cooler at −2 to 1 °C for 24 h, the following measurements were obtained: (1) cold carcass weight (CCW); (2) body wall thickness (distance between the 12th and 13th ribs beyond the ribeye, five inches from the midline of the carcass); (3) subcutaneous fat (fat thickness) was taken over the 12th to 13th thoracic vertebrae; (4) LM surface area, measured using a grid reading of the cross-sectional area of the *longissimus muscle* between 12th and 13th rib, and (5) kidney, pelvic and heart fat (KPH) was removed manually and afterward weighed and reported as a percentage of the cold carcass weight [25]. Carcass yield was estimated as (fat thickness × 0.10) + 0.40. Each carcass was split into two halves. The left side was fabricated into wholesale cuts, without trimming, according to the North American Meat Processors Association guidelines [26]. Rack, breast, shoulder and foreshank were obtained from the foresaddle, and the loins, flank and leg from the hindsaddle. Weight of each cut was subsequently recorded. The tissue composition of shoulder was assessed using physical dissection by the procedure described by Luaces et al. [27].

#### 2.1.4. Visceral Mass Data

Components of the digestive tract (GIT), including tongue, esophagus, stomach (rumen, reticulum, omasum, and abomasum), pancreas, liver, gall bladder, small intestine (duodenum, jejunum, and ileum), and large intestine (caecum, colon, and rectum) were removed and weighed. The GIT was then washed, drained, and weighed to get empty weights. The difference between full and washed digesta-free GIT was subtracted from the SBW to determine empty body weight (EBW). All tissue weights are reported on a fresh tissue basis. Organ mass is expressed as grams of fresh tissue per kilogram of final EBW, where final EBW represents the final live weight minus the total digesta weight. Full visceral mass was calculated by the summation of all visceral components (stomach complex + small intestine + large intestine + liver + lungs + heart), including digesta. The stomach complex was calculated as the digesta-free sum of the weights of the rumen, reticulum, omasum and abomasum.

#### 2.1.5. Statistical Analyses

Growth performance (ADG, DMI, and feed efficiency), estimated dietary NE and DMI, carcass data (characteristics, tissue composition, and whole cuts) and visceral mass were analyzed as a randomized complete block design, using pen as the experimental unit according to the following statistical model: Y_ij_ = μ + B_i_ + T_j_ + ε_ij_, where μ is the common experimental effect, B_i_ represents initial weight block effect, T_j_ represent dietary treatment effect, and ε_ij_ represents the residual error [28]. All the data were tested for normality using the Shapiro−Wilk test. Hot carcass weight (HCW) was used as a covariate in evaluation of treatment effects on carcass characteristics. In the analysis of shoulder tissue composition, the cold carcass weight (CCW) effect was included as a covariate. Water intake was analyzed as repeated measures using SAS PROC GLM [28]. Treatment effects were considered significant when the *p*-value was ≤0.05 and Tukey’s multiple comparison procedures were used.

### 2.2. Exp. 2. Total Tract Digestion

#### 2.2.1. Animals and Sampling

Four Pelibuey × Katahdin crossbred intact male lambs (32.7 ± 3.64 kg) were used in 4 × 4 Latin square experiment to study treatment effects on characteristics of apparent total tract digestion. Lambs were housed in individual metabolism crates (1.5 × 1.8 × 0.7 m) in an indoor facility with access to water at all times. Treatments were the same as those used in Exp. 1 (Table 1). Respective dosage of the additives (EO, EO + ENZ, VM) were hand-weighed using a precision balance (Ohaus, mod AS612, Pine Brook, NJ, USA) and top-dressed on the basal diet at the time of feeding. Chromic oxide (3.0 g/kg diet, air-dry basis) was used as an indigestible marker to estimate digestion. Chromic oxide was mixed in a 2.5-m^3^ capacity concrete mixer (model 30910-7, León Weill, SA, Coyoacán, México) for 5 min with minor ingredients (mineral supplement and urea) before being mixed with the remainder of ingredients in the basal diet. All lambs were adapted to the basal diet for 21 days before the initiation of the trial. To avoid refusals daily feed intake (as feed basis) was restricted to 1.050 kg (equivalent to the 3.2% of LW). Diets were fed in two equal proportions at 08:00 and 20:00 h daily. In order to reduce the potential for treatment carry-over effects, treatment additives were withdrawn for 7 days before initiating the next 21-day treatment period. Accordingly, experimental periods were 25 days, with 7 days of additives withdrawal (all lambs were fed the basal control diet), 18 days of adjustment to respective dietary treatments, and 3 days of sample collection. During the collection period, feces voided were collected (approximately 50 g) each day at 750, 1150 and 1550 h. Samples from each lamb and within each collection period were composited for analysis. Fecal samples were weighed, and then stored at −20 °C for subsequent analysis.

#### 2.2.2. Laboratory Analyses

Feed and fecal samples were subjected to the same analyses as feed samples of the Exp. 1, plus analysis of starch [29], gross energy (GE) by bomb calorimeter (Parr, 6400; Illinois, USA), and chromic oxide [30]. Total fecal DM excretion was estimated by the relationship of Cr intake (g) versus concentration of Cr in fecal samples as follows: total DM output, g/day = g Cr_2_O_3_ intake daily/(g Cr_2_O_3_/g of feces). Organic matter (OM) content of feed and fecal samples was estimated as DM concentration minus ash content.

#### 2.2.3. Statistical Analyses

Treatment effects on characteristics of digestion were analyzed as a 4 × 4 Latin square design following the MIXED procedure from SAS software [28], where fixed effect was treatment, and random effects were lamb and period according to the following statistical model:*Y*_*ijk*_ = µ+ *S*_*i*_ + *P*_*j*_ + *T*_*k*_ + *E*_*ijk*_
where *Y_ijk_* is the response variable, µ is the common experimental effect, *S_i_* is the lamb effect, *P_j_* is the period effect, *T_k_* is the treatment effect and *E_ijk_* is the residual error.

In all cases, least squares mean and standard error are reported. Treatment effects were tested using Fisher’s least significant difference method (LSD). Contrasts were considered significant when the *p*-value was ≤0.05, and tendencies are identified when the *p*-value was >0.05 and ≤0.10.

## 3. Results

### 3.1. Exp. 1. Growth Performance and Carcass Traits

Treatment effects on water consumption, growth performance and estimates of dietary energetics are shown in Table 2. Based on average LW and the additive dosage, dietary additive intakes averaged 3.65, 13.46, and 0.61 mg/kg LW for EO, ENZ, and VM, respectively.

Lambs that were fed the combination EO + ENZ drank 8.7% more (*p* < 0.01) water than lambs fed the other treatments (EO, VM, or with non-supplemented lambs). Water consumption for EO, VM, and control was similar (*p* > 0.66), averaging 4.47 L/d.

Growth performance and dietary energetics were not different (*p* > 0.97) for EO and VM supplemented lambs. Compared with the control, lambs supplemented with EO and VM tended (*p* = 0.09) to have greater ADG (6.3%). However, DMI was not different (*p* = 0.99). Thus, compared with control, gain-to-feed ratio (GF), observed dietary net energy (NE), and observed-to-expected diet NE were greater (*p* < 0.01) for EO and VM supplemented lambs. Compared with control, EO + ENZ increased (*p* < 0.01) 10.4% ADG. This enhancement in ADG was numerically greater than that observed with EO alone. However, EO + ENZ tended (*p* = 0.09) to increase 6.9% DMI. Compared with control EO + ENZ enhanced (*p* ≤ 0.04) in feed efficiency and observed-to-expected diet NE (3.3%). This enhancement in dietary energetic efficiency was 42% less (*p* ≤ 0.05) than the average improvement of 5.7% observed in lambs fed EO or VM. It is important to note that the observed-to-expected dietary NE and the observed-to-expected DMI ratio for the lambs fed the control diet was 0.99 (Table 2). This indicated that DMI was consistent with expectations based on observed ADG and formulated NE value of the diet (Table 1). The agreement in observed and expected DMI is supportive of the practicality of prediction equations for the estimation of DMI in relation to SBW and ADG of feedlot lambs. A dietary NE ratio (observed-to-expected dietary NE) of 1.0 is indicative that daily weight gain was consistent with observed DMI and tabular NE value of the diet taken from tables of NRC [19]. If the ratio is greater than 1, the observed dietary NE (estimated dietary NE based on growth performance) is greater than expected based on growth performance and diet formulation, indicative of enhanced metabolizable energy utilization for maintenance and gain (the reverse being the case when the ratio is less than 1).

The treatment effects on carcass characteristics, whole cuts and visceral mass, are shown in Table 3 and Table 4. Kidney-pelvic-heart fat was lower for EO vs. VM supplemented lambs (3.13 vs. 4.16%, *p* < 0.01). Kidney-pelvic-heart fat tended (*p* = 0.097) to be low for EO vs. control lambs (3.13 vs. 3.71%). However, kidney-pelvic-heart fat was not different for EO + ENZ vs. control and VM supplemented lambs. There were no treatment effects (*p* ≥ 0.37) on whole cuts or shoulder tissue composition. Compared with control, relative weight (g/kg EBW) of intestines was lower (*p* ≤ 0.03) for EO, EO + ENZ and VM supplemented lambs. This effect was more pronounced with EO supplementation. Compared to VM, supplemented EO (alone or combined with enzyme) decreased (10.1%, *p* ≤ 0.04) relative weight (g/kg EBW) of visceral fat.

### 3.2. Exp. 2. Total Tract Digestion

Treatment effects on apparent total tract digestion are shown in Table 5. There were no feed refusals. Thus, nutrient intake for all treatments were equal. Average daily gain of lambs was 0.171 kg ± 0.031 (data not shown). Thus, in this experiment, dietary additive intakes averaged 3.76, 14.04, and 0.62 mg/kg LW for EO, ENZ, and VM, respectively.

Measures of total tract digestion and digestible energy (DE; averaged 3.66 ± 0.06 Mcal DE/kg) were not different for control vs. additives. However, apparent OM digestion was greater (2.34% *p* < 0.05) for lambs supplemented with EO or EO + ENZ than for VM supplemented lambs.

## 4. Discussion

The average relative ingestion of EO + EZ in our experiment was consistent (1.01 and 1.05, respectively) to the relative ingestion observed in the experiments of Giannenas et al. [17] and Meschiatti et al. [15]. Relative ingestion of 3.5 to 4 mg EO/kg LW enhanced ruminal fermentation and feed efficiency in lactating ewes [17] and in finishing lambs [8]. Effective dosage level of ENZ for finishing lambs has not been established. However, ingestion of 12.9 mg ENZ/kg LW improved DMI and ADG in finishing steers [15]. Effects of essential oils on ruminal fermentation and growth performance are dose dependent, with enhancements being less apparent when supplemented at less than 3 mg EO/kg LW [31,32]. Daily weight gain, feed efficiency, and ruminal fermentation are enhanced when VM is supplemented within the range of 0.50 to 0.83 mg/kg LW [2,3,33]. The dosage levels of EO, ENZ and VM attained in the present study are within the reported effective ranges for enhanced growth performance and digestion.

### 4.1. Exp. 1. Growth Performance and Carcass Traits

Based on average temperature and DMI during the experiment, observed water intake for control group, EO, and VM were consistent with expected based on NRC [19], averaging 0.98, 1.01, and 0.97, respectively. The absence of effect of EO and VM on water intake has been reported previously [34,35]. In contrast, compared with the other treatments, supplementation with EO + ENZ increased (7%) water consumption. Likewise, Valdés et al. [36] observed increased (6.4%) water consumption with EO + ENZ supplementation. Consistent with the present study, the increase in water intake was largely due to increased DMI. Indeed, water consumption/kg DMI was similar across treatments, averaging 6.80, 6.98, 6.92, and 6.76 for control, EO, EO + ENZ, and VM, respectively.

Information regarding the effects of the standardized mixture of essential oils used in the present study (CRINA Ruminants) on growth performance and dietary energetics in finishing cattle and lambs fed with high-energy diets is limited. Meyer et al. [6] observed a 4% increase in efficiency of dietary energy utilization in finishing cattle fed a high-energy corn-based diet (2.20 Mcal NE_m_/kg diet DM). Likewise, Plascencia et al. [8] observed that compared with control, EO supplementation enhanced both feed efficiency (4.7%) and estimated dietary net energy (3.1%) in fattening lambs fed with a cracked corn-based diet (2.14 Mcal NE_m_/kg diet DM). The basis for enhanced efficiency of energy utilization with EO supplementation has not been established. Supplemental EO has altered ruminal VFA molar ratios, with increased proportion of propionate and decreased methane production [6,17,36].

In finishing lambs, GF responses to supplemental VM have been variable, ranging from nil (lambs fed a low-to-moderate energy diet (~1.80 Mcal NE_m_/kg diet DM) with a dose of 17.9 mg VM/kg diet [37]) up to a 10% increase (lambs fed a moderate energy diet (~1.95 Mcal NE_m_/kg diet DM) with a dose of 20 mg/kg [38]). Growth performance response to supplemental VM depends on both dosage level and dietary energy density [3]. Consistent with the present study, enhancements of 3 to 9% in ADG and 4 to 11% in feed efficiency have been reported for feedlot cattle fed high-energy corn-based diets (>2.17 Mcal NE_m_/kg diet DM) supplemented with 22 to 26 mg VM/kg diet DM [2,39]. The positive effect of VM on growth performance has been associated with enhanced N utilization and reduced liver abscess incidence [39,40].

The increased ADG with EO + ENZ supplementation observed in the present experiment was due to enhanced DMI. Indeed, the efficiency of energy utilization (expected vs. observed dietary NE) was appreciably lower (2.5%) for EO + ENZ vs. EO alone. The basis for the slightly lower energetic efficiency for EO + ENZ is not certain, but may be due to changes in site of starch digestion (increased ruminal vs. postruminal digestion; Owens et al. [41]). Tricarico et al. [42] reported results of four experiments evaluating α-amylase supplementation of feedlot cattle. Consistent with the present study, they observed supplemental α-amylase increased DMI by an average of 5.2%. Meschiatti et al. [7] observed a 2.3% increase in DMI of feedlot steers supplemented with the combination of α-amylase (560 mg/kg diet DM) and essential oils (90 mg /kg diet DM). Valdés et al. [36] observed an 11% increase in DMI and a 22% increase in ADG of lambs fed a maize silage-based diet (70% maize silage and 30% concentrate) supplemented with an enzyme mixture alone (α-amylase, endoglucanase, and xylanase). The effect was further enhanced when fed in combination with essential oils (blend of salicin, myricetin, kaempferol, and quercetin). Klingerman et al. [14] observed a 5% increase in DMI of lactating cows. Responses to supplemental α-amylase may be greatest in livestock that are fed diets containing otherwise less digestible corn hybrids (i.e., flinty corn with high concentrations of vitreous endosperm).

Lack of treatment effects on carcass traits and tissue composition is consistent with previous comparisons of supplemental EO [8,43], EO + ENZ combination [44], and VM [39].

The decrease in intestinal mass with VM supplementation is consistent with studies in which VM supplementation decreased intestinal wall thickness, and hence, intestinal weight of mice [45], broilers [46], and cattle [47]. Likewise, we observed decreased intestinal mass with EO supplementation, consistent with antibiotic-like effects on intestinal epithelial thickness. Wang et al. [48] observed that supplemental essential oils (mainly thymol) decreased jejunal wall thickness of poultry, while Ghazanfari et al. [49] observed that supplementation with essential oil mixture (mixture of linalool, terpinene, and limonene) decreased wall thickness along all segments of the small intestine of poultry by an average of 30%.

The effects of supplemental EO and EO + ENZ vs. VM on mesenteric and visceral fat (g/kg EBW) is uncertain. It has been proposed that supplemental EO may have potential as an energy “repartitioning” agent, affecting net fat deposition and distribution [50]. This may partially explain changes in meat quality of lambs supplemented with EO [51,52]. To the extent that EO reduces ruminal acetate: propionate ratio [6,43], the associated increase in propionate production may lead to decreased visceral fat deposition [53].

### 4.2. Exp. 2. Total Tract Digestion

For the most part, supplemental EO has not appreciably affected measures of apparent total tract digestion in dairy cattle [31,54], steers [9,55,56] and in lambs [10,57,58]. In several studies [54,57,59], supplemental essential oils were found to alter ruminal fermentation without effect on total tract digestion.

Consistent with Gouvêa et al. [44], the combination EO + ENZ did not affect apparent total tract digestion of OM, starch or NDF. Lack of treatment effects on total tract starch digestion is not surprising, in as much as apparent total tract starch digestion approached 100% for all treatments. The completeness of starch digestion in our study is consistent with numerous prior studies evaluating starch digestion in feedlot lambs fed high-grain diets [60,61,62]. Other researchers noted that whereas amylase supplementation of lambs did not affect total tract starch digestion, it increased ruminal starch digestion [63]. Ruminal digestion of cracked corn in lambs averages 77% [64,65]. Applying the above to the daily starch intake in the present study (499 g), the estimated ruminal escape of starch to the small intestine was 115 g/d. Xu et al. [65] observed that maximum intestinal α-amylase activity is reached when rumen escape starch is about 100–120 g/d in 25–30 kg lambs. Klingerman et al. [14] observed that although supplemental amylase did not affect total tract starch digestion, it increased NDF digestion in cows fed a high-forage diet.

Supplemental VM did not affect measures of total tract digestion. In a series of experiments evaluating the effects of 0 vs. 25 mg VM/kg DM fed to feedlot lambs, Fiems et al. [66] observed that although VM did not affect apparent total tract OM digestion, it decreased apparent total tract digestion of CP and ether extract. Da Fonseca et al. [67] did not observe any effect of VM supplementation (30 mg/kg DM) on apparent total tract digestion of DM, OM, N, and NDF in steers fed 50:50 concentrate: forage diet. Salinas et al. [2] did not observe effects on measures of total tract digestion in steers receiving a steam-flaked corn-based finishing diet supplemented with 22.5 mg VM/kg DM. Feeding greater dosage levels of VM (26 to 28 mg VM/kg DM) in feedlot steers fed similar finishing diets likewise did not affect apparent total tract digestibility of OM, NDF, starch, and CP [3,39]. The lower OM digestion observed for lambs receiving VM treatment vs. control, EO and EO + ENZ treatments is not certain. In previous reports [2,3,39], supplemental VM resulted in numerically lower apparent total tract OM digestion (averaging −2.6%) than that of control steers. In the present study, average difference between EO and EO + ENZ vs. VM was 2.10%. Since there were no effects of additives on total tract digestion, digestible energy (DE) of the diet were similar (*p* ≥ 0.28) between treatments, averaging 3.66 ± 0.06 Mcal NE_m_/kg. Applying the relationship between ED and NE_m_ derived by Zinn and Plascencia (NE_m_ = 0.736DE − 0.661) [68], the NE_m_ value in digestion trial resulted in 2.03 Mcal NE_m_/kg; this is in close agreement (0.975) with the value of 2.08 Mcal NE_m_/kg observed for the basal diet in the growth performance trial (Exp. 1).

## 5. Conclusions

Supplemental EO may be an effective alternative to VM in high-energy finishing diets for feedlot lambs, enhancing both feed efficiency and apparent efficiency of energy utilization. Combination EO + ENZ may further enhance dry matter intake, promoting increased weight gain.

## Figures and Tables

**Table 1 animals-11-02390-t001:** Composition of basal diet fed by lambs in Exp. 1 and Exp. 2 ^†^.

		Treatments ^§^		
Item	Control	EO	EO + ENZ	VM
Ingredient composition, % DM basis				
Dry-rolled corn	67.00	67.00	67.00	67.00
Sudangrass hay	9.00	9.00	9.00	9.00
Soybean meal	10.00	10.00	10.00	10.00
CRINA-Ruminants^®^	0	+++	0	0
RONOZYME Rumistar^®^	0	0	+++	0
Stafac 500^®^	0	0	0	+++
Molasses cane	9.00	9.00	9.00	9.00
Urea	0.50	0.50	0.50	0.50
Tallow	2.00	2.00	2.00	2.00
Trace mineral salt *	2.50	2.50	2.50	2.50
Chemical composition (%DM basis) ^‡^				
Dry matter	88.60	88.60	88.60	88.60
Starch	54.00	54.00	54.00	54.00
Neutral detergent fiber	13.50	13.50	13.50	13.50
Crude protein	13.66	13.66	13.66	13.66
Ether extract	5.10	5.10	5.10	5.10
Calculated net energy (Mcal/kg) ^⁋^				
Maintenance	2.08	2.08	2.08	2.08
Gain	1.43	1.43	1.43	1.43

^†^ Chromic oxide (0.30%) was added, replacing dry-rolled corn, as digesta marker in Exp. 2. ^§^ Control = non-supplemented; doses per lamb, EO = a standardized source of a mixture of essential oils compounds at dose of 150 mg EO (CRINA^®^Ruminants, DSM Nutritional Products, Basel, Switzerland)/day; EO + ENZ = 150 mg EOC plus 560 mg alpha-amylase (RONOZYME Rumistar^®^, DSM Nutritional Products, Basel, Switzerland)/day; VM = a peptolide antibiotic virginiamycin (Stafac 500, Phibro Animal Health, Ridgefield Park, NJ) at dose of 25 mg virginiamycin/day. * Mineral premix contained: limestone, 50%; urea 20%; NaCl, 15%; MgO, 5%; phosphate rock, 9.06%; CoSO4, 0.01%; CuSO4, 0.14%; FeSO4, 0.47%; ZnO, 0.16%; MnSO4, 0.14%; KI, 0.008%. ^‡^ Average dietary composition was determined by analyzing subsamples collected and composited throughout the experiment. Accuracy was ensured by adequate replication with acceptance of mean values that were within 5% of each other. ^⁋^ Calculated from NRC [19] tabular values.

**Table 2 animals-11-02390-t002:** Effect of treatments on growth performance of finishing lambs.

	Treatments ^†^							*p*-Value			
Item	Control	EO	EO + ENZ	VM	SEM	1 vs. 2	1 vs. 3	1 vs. 4	2 vs. 3	2 vs. 4	3 vs. 4
Days on test	87	87	87	87							
Pen replicates	6	6	6	6							
Water intake, L/d	4.44	4.56	4.85	4.40	0.174	0.56	<0.01	0.25	<0.01	0.15	0.01
Live weight, kg/d ^§^											
Initial	27.88	27.89	27.87	27.82	0.104	0.90	0.96	0.70	0.86	0.62	0.75
Final	52.49	54.16	55.31	54.14	0.623	0.09	<0.01	0.10	0.27	0.98	0.61
Average daily gain, kg/d	0.283	0.302	0.316	0.302	0.007	0.09	<0.01	0.09	0.21	0.99	0.21
Dry matter intake, kg/d	1.305	1.306	1.401	1.301	0.037	0.99	0.09	0.92	0.09	0.93	0.07
Feed efficiency, kg/kg	0.217	0.232	0.227	0.233	0.002	<0.01	0.02	<0.01	0.13	0.85	0.10
Diet net energy, Mcal/kg											
Maintenance	2.08	2.19	2.14	2.20	0.017	<0.01	0.04	<0.01	0.06	0.83	0.04
Gain	1.42	1.51	1.47	1.52	0.015	<0.01	0.04	<0.01	0.06	0.83	0.04
Observed-to-expected diet NE, Mcal/kg											
Maintenance	1.001	1.053	1.027	1.056	0.009	<0.01	0.04	<0.01	0.06	0.83	0.04
Gain	0.991	1.052	1.024	1.055	0.011	<0.01	0.04	<0.01	0.06	0.83	0.04
Observed-to-expected DMI	1.006	0.948	0.976	0.944	0.010	<0.01	0.04	<0.01	0.06	0.83	0.04

^†^ Control = non-supplemented; doses per lamb, EO = a standardized source of a mixture of essential oils compounds at dose of 150 mg EO (CRINA^®^Ruminants, DSM Nutritional Products, Basel, Switzerland)/day; EO + ENZ = 150 mg EOC plus 560 mg alpha-amylase (RONOZYME Rumistar^®^, DSM Nutritional Products, Basel, Switzerland)/day; VM = a peptolide antibiotic virginiamycin (Stafac 500, Phibro Animal Health, Ridgefield Park, NJ) at dose of 25 mg virginiamycin/day.

**Table 3 animals-11-02390-t003:** Effect of treatments on carcass characteristics of finishing lambs.

	Treatments ^†^							*p*-Value			
Item	Control	EO	EO + ENZ	VM	SEM	1 vs. 2	1 vs. 3	1 vs. 4	2 vs. 3	2 vs. 4	3 vs. 4
Hot carcass weight, kg	31.24	32.68	32.93	32.67	0.73	0.18	0.12	0.18	0.81	0.99	0.80
Dressing percentage	59.47	60.30	59.48	60.33	0.83	0.19	0.98	0.47	0.48	0.98	0.47
Cold carcass weight, kg	30.93	32.28	32.63	32.28	0.74	0.21	0.40	0.58	0.99	0.99	0.99
LM area, cm^2^	18.98	19.58	19.55	18.78	0.49	0.39	0.42	0.78	0.97	0.27	0.29
Fat thickness, cm ^§^	0.283	0.256	0.253	0.262	0.19	0.32	0.27	0.43	0.91	0.83	0.75
Kidney pelvic and heart fat, %	3.72	3.13	3.65	4.16	0.25	0.09	0.83	0.21	0.14	<0.01	0.15
Carcass yield *	1.52	1.41	1.40	1.43	0.07	0.31	0.27	0.42	0.93	0.83	0.76
Shoulder composition, %											
Muscle	64.86	63.72	63.64	63.60	0.55	0.17	0.14	0.13	0.92	0.88	0.97
Fat	16.22	16.99	17.39	17.32	0.60	0.37	0.18	0.25	0.64	0.79	0.84
Muscle to fat ratio	4.03	3.80	3.68	3.72	0.16	0.30	0.15	0.20	0.66	0.78	0.87
Whole cuts (as percentage of CCW)											
Forequarter	41.66	41.74	41.69	41.65	0.40	0.88	0.96	0.99	0.92	0.87	0.95
Hindquarter	37.32	38.03	37.56	37.41	0.43	0.26	0.70	0.88	0.45	0.33	0.82
Neck	7.53	7.99	7.73	7.68	0.26	0.23	0.59	0.69	0.49	0.41	0.89
Shoulder IMPS206	14.12	13.93	14.06	13.90	0.19	0.48	0.81	0.42	0.65	0.92	0.57
Shoulder IMPS207	9.42	8.97	9.18	9.38	0.23	0.18	0.47	0.89	0.53	0.23	0.56
Rack IMPS204	5.87	6.05	5.92	6.11	0.13	0.36	0.81	0.24	0.50	0.77	0.34
Breast IMPS209	5.61	5.57	5.24	5.59	0.16	0.86	0.12	0.93	0.15	0.93	0.13
Ribs IMPS209A	6.52	6.71	6.66	6.72	0.15	0.37	0.53	0.36	0.78	0.99	0.77
Loin IMPS231	7.11	7.16	7.00	7.02	0.18	0.85	0.69	0.73	0.56	0.60	0.96
Flank IMPS232	6.09	6.23	6.30	6.26	0.21	0.63	0.49	0.57	0.83	0.93	0.89
Leg IMPS233	24.39	24.58	24.16	24.08	0.33	0.68	0.62	0.51	0.37	0.29	0.86

^†^ Control = non-supplemented; doses per lamb, EO = a standardized source of a mixture of essential oils compounds at dose of 150 mg EO (CRINA^®^Ruminants, DSM Nutritional Products, Basel, Switzerland)/day; EO + ENZ = 150 mg EOC plus 560 mg alpha-amylase (RONOZYME Rumistar^®^, DSM Nutritional Products, Basel, Switzerland)/day; VM = a peptolide antibiotic virginiamycin (Stafac 500, Phibro Animal Health, Ridgefield Park, NJ) at dose of 25 mg virginiamycin/day. ^§^ Fat thickness over the center of the LM between the 12th and 13th ribs. * Carcass yield was estimated as (fat thickness × 0.10) + 0.40.

**Table 4 animals-11-02390-t004:** Effect of treatments on visceral mass of finishing lambs.

	Treatments ^†^							*p*-Value			
Item	Control	EO	EO + ENZ	VM	SEM	1 vs. 2	1 vs. 3	1 vs. 4	2 vs. 3	2 vs. 4	3 vs. 4
GIT fill, kg	3.93	4.00	4.38	3.84	0.26	0.86	0.21	0.80	0.28	0.67	0.14
Empty body weight, % of full weight	92.50	92.58	92.00	92.88	0.49	0.90	0.48	0.58	0.41	0.67	0.22
Full viscera, kg	7.83	7.86	8.45	7.79	0.28	0.93	0.13	0.92	0.16	0.86	0.12
Organs, g/kg of empty body weight											
Stomach complex	26.22	26.52	27.16	26.81	0.79	0.79	0.41	0.60	0.57	0.79	0.76
Intestines	53.97	50.49	52.27	51.71	0.48	<0.01	0.03	<0.01	0.02	0.10	0.42
Liver/spleen	16.79	15.74	15.86	16.08	0.43	0.11	0.16	0.27	0.82	0.57	0.73
Heart/lungs	20.86	19.73	19.71	19.97	0.66	0.17	0.11	0.28	0.49	0.75	0.32
Kidney	2.42	2.34	2.55	2.35	0.09	0.37	0.17	0.43	0.12	0.92	0.15
Omental fat	32.05	29.51	30.89	32.22	0.96	0.08	0.40	0.91	0.32	0.06	0.34
Mesenteric fat	11.20	10.92	11.07	13.58	0.73	0.79	0.90	0.04	0.88	0.02	0.03
Visceral fat	43.26	40.39	41.96	45.81	1.16	0.11	0.44	0.14	0.36	0.01	0.04

^†^ EO = a standardized source of a mixture of essential oils compounds at dose of 150 mg EO/kg diet DM (CRINA^®^Ruminants, DSM Nutritional Products, Basel, Switzerland); EOC + ENZ = 150 mg EO plus 560 mg alpha-amylase (RONOZYME Rumistar^®^, DSM Nutritional Products, Basel, Switzerland)/kg DM; VM = a peptolide antibiotic virginiamycin (VM, Stafac 500 Phibro Animal Health, Ridgefield Park, NJ) at dose of 25 mg virginiamycin/kg DM.

**Table 5 animals-11-02390-t005:** Effect of treatments on nutrient digestion.

	Treatments ^†^							*p*-Value			
Item	Control	EO	EO + ENZ	VM	SEM	1 vs. 2	1 vs. 3	1 vs. 4	2 vs. 3	2 vs. 4	3 vs. 4
Intake, g/d											
Dry matter	924	924	924	924							
Organic matter	866	866	866	866							
Starch	499	499	499	499							
NDF	124.7	124.7	124.7	124.7							
N	20.20	20.20	20.20	20.20							
Ether extract	47.12	47.12	47.12	47.12							
Gross energy, Mcal/d	4.055	4.055	4.055	4.055							
Fecal excretion, g/d											
Dry matter	168.0	160.8	163.1	177.3	4.35	0.28	0.45	0.18	0.72	0.04	0.06
Organic matter	143.6	137.1	138.5	154.0	5.44	0.28	0.39	0.17	0.81	0.03	0.05
Starch	3.66	2.58	3.75	3.91	0.52	0.20	0.90	0.74	0.16	0.12	0.84
NDF	60.96	58.30	59.81	63.44	3.06	0.56	0.80	0.59	0.74	0.28	0.43
N	4.69	4.61	4.76	4.51	0.15	0.68	0.77	0.40	0.49	0.66	0.27
Ether extract	8.91	9.77	9.35	8.57	1.18	0.50	0.73	0.78	0.73	0.35	0.54
Gross energy, Mcal/d	0.685	0.645	0.648	0.696	0.074	0.71	0.73	0.92	0.97	0.64	0.66
Total tract digestion, %											
Dry matter	81.94	82.76	82.37	80.79	0.71	0.29	0.57	0.16	0.60	0.04	0.07
Organic matter	83.54	84.33	84.02	82.21	0.45	0.28	0.49	0.15	0.66	0.03	0.06
Starch	99.29	99.48	99.23	99.22	0.15	0.24	0.71	0.66	0.14	0.13	0.95
NDF	51.18	53.66	52.35	49.17	2.32	0.48	0.73	0.56	0.70	0.22	0.37
N	76.99	77.26	76.42	77.73	0.76	0.81	0.62	0.52	0.47	0.68	0.27
Ether extract	82.86	81.60	82.39	83.47	0.23	0.61	0.85	0.80	0.75	0.46	0.66
Digestible energy, %	83.22	84.26	84.07	82.60	1.06	0.36	0.46	0.70	0.86	0.22	0.28
Digestible energy, cal/kg	3.65	3.70	3.69	3.63	0.033	0.36	0.46	0.70	0.86	0.22	0.28

^†^ Control = non-supplemented; doses per lamb, EO = a standardized source of a mixture of essential oils compounds at dose of 150 mg EO (CRINA^®^Ruminants, DSM Nutritional Products, Basel, Switzerland)/day; EO + ENZ = 150 mg EOC plus 560 mg alpha-amylase (RONOZYME Rumistar^®^, DSM Nutritional Products, Basel, Switzerland)/day; VM = a peptolide antibiotic virginiamycin (Stafac 500, Phibro Animal Health, Ridgefield Park, NJ) at dose of 25 mg virginiamycin/day.

## Data Availability

Not applicable.

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
