# Peer review of "Blend of Essential Oils Supplemented Alone or Combined with Exogenous Amylase Compared with Virginiamycin Supplementation on Finishing Lambs: Performance, Dietary Energetics, Carcass Traits, and Nutrient Digestion"

_animals, 2021, doi:10.3390/ani11082390_

Round 1
Reviewer 1 Report
It is an article that evaluates the use of zootechnical additives in the diet of lambs fed with high starch content. The results, although not very innovative, show the joint effect of a blend of essential oil and amylase. The authors need to clarify some points better in the introduction and in the material and methods. Please see the attached material.

Author Response
We are grateful to reviewers for the time and effort in helping improve the quality of the manuscript. The observations were wise and timely which permit the improvement substantially the manuscript.
All changes and correction made are highlighted (tracked) in yellow in the corrected version of the manuscript.
For general and specific responses please see the attachment

Reviewer 2 Report
The research study by Alfredo Estrada-Angulo et al., entitled ‘Blend of essential oils supplemented alone or combined with exogenous amylase compared with virginiamycin supplementation on finishing lambs: Performance, dietary energetics, carcass traits, and nutrient digestion’ appears to be original and the information in the paper is suitable for Animals readers. Although authors have undertaken a serious research effort yet there are certain issues that should be thoroughly addressed. The statistical analysis needs to be appropriate to compare the effects of different tretments and the P-values should be provided in the tables. The calculation of increase or decrease in mean values in treatment groups against control needs to be rechecked.
The information in Material and methods is provided in details.
Discussion and interpretations of findings is done quite well except for clarity in some sentences.
Please have the whole paper checked thoroughly and corrected by an English-speaking native speaker who is familiar with scientific writing. The grammar would benefit from corrective action as in places the text does not read particularly well.
Specific comments are given directly in the manuscript and the author should use Adobe acrobat Reader DC to see the comments indicated in the manuscript.
The manuscript in the current form is not acceptable and a major revision is recommended.

Author Response

(The authors gave the same response as above.)
